# Leaf Count Aided Novel Framework for Rice (*Oryza sativa* L.) Genotypes Discrimination in Phenomics: Leveraging Computer Vision and Deep Learning Applications

**DOI:** 10.3390/plants11192663

**Published:** 2022-10-10

**Authors:** Mukesh Kumar Vishal, Rohit Saluja, Devarshi Aggrawal, Biplab Banerjee, Dhandapani Raju, Sudhir Kumar, Viswanathan Chinnusamy, Rabi Narayan Sahoo, Jagarlapudi Adinarayana

**Affiliations:** 1CSRE, Indian Institute of Technology Bombay, Mumbai 400076, India; 2CSE, Indian Institute of Technology Bombay, Mumbai 400076, India; 3Indian Institute of Information Technology, Hyderabad 500032, India; 4Indian Council of Agricultural Research—Indian Agricultural Research Institute, Pusa, New Delhi 110012, India

**Keywords:** number of leaves, biomass, deep learning, genome wide association study (GWAS), high throughput plant phenotyping (HTPP), leaf counting, rice (*Oryza sativa* L.), phenomics

## Abstract

Drought is a detrimental factor to gaining higher yields in rice (*Oryza sativa* L.), especially amid the rising occurrence of drought across the globe. To combat this situation, it is essential to develop novel drought-resilient varieties. Therefore, screening of drought-adaptive genotypes is required with high precision and high throughput. In contemporary emerging science, high throughput plant phenotyping (HTPP) is a crucial technology that attempts to break the bottleneck of traditional phenotyping. In traditional phenotyping, screening significant genotypes is a tedious task and prone to human error while measuring various plant traits. In contrast, owing to the potential advantage of HTPP over traditional phenotyping, image-based traits, also known as i-traits, were used in our study to discriminate 110 genotypes grown for genome-wide association study experiments under controlled (well-watered), and drought-stress (limited water) conditions, under a phenomics experiment in a controlled environment with RGB images. Our proposed framework non-destructively estimated drought-adaptive plant traits from the images, such as the number of leaves, convex hull, plant–aspect ratio (plant spread), and similarly associated geometrical and morphological traits for analyzing and discriminating genotypes. The results showed that a single trait, the number of leaves, can also be used for discriminating genotypes. This critical drought-adaptive trait was associated with plant size, architecture, and biomass. In this work, the number of leaves and other characteristics were estimated non-destructively from top view images of the rice plant for each genotype. The estimation of the number of leaves for each rice plant was conducted with the deep learning model, YOLO (You Only Look Once). The leaves were counted by detecting corresponding visible leaf tips in the rice plant. The detection accuracy was 86–92% for dense to moderate spread large plants, and 98% for sparse spread small plants. With this framework, the susceptible genotypes (MTU1010, PUSA-1121 and similar genotypes) and drought-resistant genotypes (Heera, Anjali, Dular and similar genotypes) were grouped in the core set with a respective group of drought-susceptible and drought-tolerant genotypes based on the number of leaves, and the leaves’ emergence during the peak drought-stress period. Moreover, it was found that the number of leaves was significantly associated with other pertinent morphological, physiological and geometrical traits. Other geometrical traits were measured from the RGB images with the help of computer vision.

## 1. Introduction

In recent times, plant phenomics and high throughput plant phenotyping (HTPP) technologies have evolved as an efficient way for studying plants and alleviating the bottleneck of traditional phenotyping for precision agriculture [1,2,3,4,5,6]. These non-destructive technologies employ an array of sensors for imaging the plant, remote sensing principles, state-of-art image analysis techniques aided with computer vision, AI-ML, and deep learning algorithms, in order to estimate plant traits associated with the physiology, morphology, geometry, and to observe plant experiments under laboratory and field conditions [1,7,8,9,10]. These technologies have many integral advantages such as being fast, efficient, cost-effective, non-invasive, and offering uninterrupted monitoring of an experiment in an automatic to semi-automatic method around the clock [2,6]. Owing to these potential advantages, they have recently been intensively adopted in the genetic improvement program, plant phenotyping, and genome-wide association studies (GWAS) of crops, with the primary aim of decoding biotic and abiotic stress tolerance functionality with their associated regulatory genes [3,5,6,11,12,13,14,15].

Across the globe, a large number of ongoing research work targets the development of climate-resilient novel high-yielding varieties, especially varieties that are tolerant or resistant to biotic and abiotic stress, for food security. The ever-burgeoning population of the planet has posed the challenge of at least doubling food and fiber production, or increasing it manyfold in the future, with available limited genetics and natural resources [11,16,17,18,19]. Furthermore, enhancing cereal food production is a prime concern, as it is a staple food worldwide. Among cereal crops, rice (*Oryza sativa* L.) is an essential crop that feeds a significant amount of the population worldwide [16].

Rice is a major staple crop in India and other populous developing nations. Rice is also a water-intensive crop; thus, amid depleting groundwater scenarios and increased occurrence of drought, we need water-efficient high yielding cultivars for rice [11,16,18], particularly, drought-tolerant genetics to extenuate such issues [11,20]. It is well-known that drought is a prime detrimental factor to gaining a high yield of rice. Therefore, efforts are being made to discover drought-resistant new varieties and genetic traits by phenomics and HTPP, in addition to traditional phenotyping methods.

Many studies report that rice plant architecture is a definitive factor in achieving high grain productivity [21,22,23]. In rice, plant architecture plays a vital role, in particular, the crucial traits of number of leaves, plant height, their spatial pattern, and the number of tillers [16,21,22,24]. These traits are directly associated with biomass and yield. In comparison with traditional phenotyping, with the help of phenomics and HTPP, it is possible to dive more deeply into the plant architecture of rice and estimate associated traits using plant images, often termed by many research works, as image-traits, i-traits or digital features [1,9,14,25,26,27].

These image traits can be employed to understand plants, their behavior, genetic make-up, and identify new image-based traits of plants and their linked genes. These image traits can further be utilized for discriminating the plant’s genotype, monitoring plants in an experiment, breeding programs, and for GWAS studies [6,13,14,28]. In GWAS, the main task is to decode genotype–phenotype association and discover the responsible gene(s) associated with a specific trait [6,13]. Such studies investigate genetic markers that can predict the presence of a trait across the whole genome of a population with the help of statistical analysis. In a recent GWAS study in rice utilizing i-trait, it was reported that around 94% of loci were related to drought resistance (DR) [14]. They found that the OsPP15 gene was associated with drought in rice [14]. In other phenomics-based GWAS analyses, drought-resistant associated genes, such as GhRD2, GhNAC4, GhHAT22, and GhDREB2, were identified for cotton with the help of image traits [27]. Therefore, image traits are increasingly providing more information and facilitating discovery of physiological traits and associated genes. Further experimental studies are needed under laboratory and field conditions to validate the estimation of image-based-traits and their association with different genes for different crops. Subsequently, large experiments on this subject will ascertain the role of image traits in dissecting regulatory genes for linked traits [2,6,12].

Additionally, these features have been employed in clustering genotypes based on phenotypic plant traits. They have been used in breeding programs to estimate morphological and physiological traits, and to more deeply understand the phenotype [6,13]. Beyond this, phenomics and HTTP have been employed to understand the gene and QTL in many GWAS studies [5,6,13,14,27]. HTPP is a platform that employs extensive image processing, computer vision, machine learning, and deep learning algorithms for such studies. The newer and more powerful AI algorithms make it possible to measure image traits effortlessly, with higher accuracy and precision, over traditionally measured characteristics in an experiment [6,29].

Our present study attempts to estimate a few such image-based phenotypic traits, particularly the number of leaves and leaf emergence rate under drought conditions, to understand genotypes and discriminate between the phenotypes in a GWAS study. The current work includes an extended piece of leaf counting utilizing the You Only Look Once (YOLO) algorithm [30]. This crucial trait was employed to find a correlation with other geometrical–morphological traits associated with plant size and biomass. The current work increased the total training and testing image samples for counting leaves for the models. In earlier work, 450 RGB images were taken [30]. In this work, 750 rice plant images were employed to label around 25,000 leaf tips to achieve higher counting accuracy and precision. To validate the result, we additionally used data other than the GWAS 2018 experiment, i.e., ‘Nitrogen Use Efficiency (NUE) 2019’ and ‘Water Use Efficiency (WUE) 2020’ rice experiments conducted at the Indian Council of Agricultural Research–Indian Agricultural Research Institute, ICAR-IARI, for training and testing samples.

Moreover, the leaf counting hypothesis was the same: each leaf has a single tip; thus, we can calculate leaf numbers in a plant by counting the associated tips. Past works confirm that the number of leaves is directly associated with other image traits, the plant size, plant architecture, and, thus, the biomass [16,21,30]. With predicted tips and their coordinates, other i-traits such as the convex hull area, bounding box area, the number of leaves per convex hull area, and the plant aspect ratio, were estimated. This is an advantage of our proposed framework. In YOLO, an object is predicted with a bounding box around it [31]. Here, we detected and indicated the leaf tip as an object with a bounding box around it with its tip coordinate. The YOLO algorithm was preferred owing to its speed of detecting a real-time object [31].

Finally, we grouped the diverse genotypes based on the predicted number of leaves found for each genotype under drought and control conditions as drought-adaptive image traits. We estimated the number of leaves emerging during the induced stress period for genotypes, and compared these diverse genotypes to discriminate for drought-tolerant and drought-sensitive genotype groups.

## 2. Materials and Methods

### 2.1. The Experimental Set-Up and Details

The entire experiment was set at the Nanaji Deshmukh Plant Phenomics Center (NDPPC) at the Indian Council of Agricultural Research (ICAR)–Indian Agricultural Research Institute (IARI) located in New Delhi, India, 28.6331° N, 77.1525° E. This phenotyping facility was set up by LemnaTec GmbH, Aachen, Germany. NDPPC has an automated phenotyping platform of plant-to-sensor type [2], where plants move on conveyors to an imagining station, Figure 1a. The imaging chamber has automatic sensors such as RGB, thermal, NIR, SWIR, VNIR, and fluorescence. For the current experiment, RGB images of rice plants were captured by the RGB sensor, model Prosilica GT 6600 series.

This phenomics center has four climate-controlled greenhouses where the plants were grown, Figure 1b,c. All rice genotypes were transplanted with four replications in the 2018 experiment GWAS. The replication process of each genotype removes bias, improves the confidence of the result, and provides a better error estimate. It also increases the validation of the result while conducting the phenotyping experiments [32]. However, additional 2019 and 2020 experiment image datasets were also taken randomly as a subset of images for training and testing purposes for the deep learning model.

### 2.2. The Datasets

This work used IARI RICE GWAS (Genome-Wide Association Study) 2018. However, for training the model for leaf counting, three datasets— “IARI RICE GWAS (Genome-Wide Association Study) 2018”, “IARI RICE NUE (Nitrogen Use Efficiency) 2019”, and “IARI RICE WUE (Water Use Efficiency) 2020”—were used. “IARI RICE GWAS 2018” is a dataset acquired from diverse rice genotypes for GWAS analysis in 2018. “IARI RICE NUE 2019” and “IARI RICE WUE 2020” are datasets that arose from the nitrogen use efficiency (NUE) experiment carried out on rice in 2019 and Water Use Efficiency (WUE) experiment for the recombinant inbred line (RIL) population in 2020, respectively. The dataset of our study contains RGB images comprising a top view and side view of each rice plant, Figure 2. Each RGB image has a resolution of 6576 × 4384 for the top view, and 4384 × 6576 for the side view. These three datasets were employed to create a subset of the sample to train the deep learning algorithms for leaf counting, consisting of diverse images of the genotypes in the datasets, as deep learning algorithms are data-hungry.

However, this paper presents the phenotyping and image analysis results of the “IARI RICE GWAS 2018” dataset to analyze and understand the genotype images. The 110 diverse genotypes of the “IARI RICE GWAS 2018” experiment was used for this work. We estimated these phenotypes’ image traits (i-traits) in the present investigation. We suggested new i-traits such as ‘the number of leaves per convex hull area’ as associated with the plant size and biomass, leaf distribution pattern type, and for quantification of drought stress [33]. The side view (SV) image data for each plant was comprised of four side views—0°, 90°, 120° and 240°—to capture a greater variety of information about each plant. However, we used RGB images—“Top View (TV) and Side View (SV)—0°, 90°” for the analysis, results, and discussions.

**Figure 2 plants-11-02663-f002:**
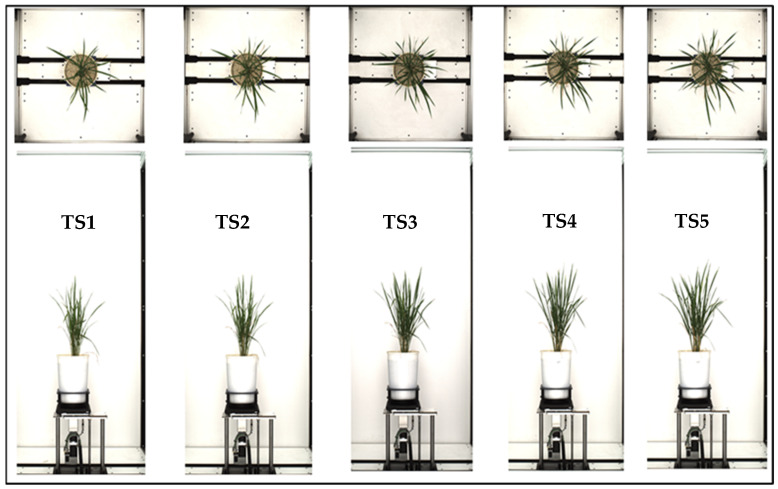
Rice plants images: side view (SV) and top view (TV) at different growth stages; TS—time snap, after inducing drought stress. TS-4 is the peak drought stress, and TS5 is after rewatering of the plants.

### 2.3. Plant Materials and Growing Conditions

As mentioned previously, a total of 110 diverse rice genotypes were examined in the GWAS experiment, 2018. These genotypes included mega Indian varieties (admixtures), germplasm lines from Aus, Indica, and Japonica origin species, and the Bioprospecting and Allele Mining (BAM) Project Mini-Core rice collection, mostly from Asia. The entire experiment was conducted during the *Kharif* (rainy, southwest monsoon) season, which is the rice cultivation period in India. Initially, all genotypes of rice seeds were sown at the experimental field of NDPPC, and at 30 days old, seedlings were transplanted to a plastic pot containing 15 kg of puddled fertile soil. The plastic pot was manufactured from white virgin plastic, LDPE, with dimensions of 24 cm (diameter at the neck, top of the pot) × 47 cm (height) × 19 cm (diameter at bottom) and approximately twenty liters by volume. To mimic local field conditions, each pot was filled with the local field soil of IARI. The soil was dominant in sandy loam types. Fifteen days after transplanting, pots of uniformly grown plants of each genotype were selected and placed on conveyor belts installed at climate-controlled greenhouses for high throughput plant phenotyping. After an adaptation period of 24 days inside greenhouse conditions, imaging was performed for a five-time snap (TS5).

Common agricultural practice (120N:60P:80K) was conducted for all replicated pots, and soil moisture content (SMC) was maintained up to saturation at 25% *w*/*v* using an automatic gravimetric scale. Each pot was irrigated twice a day before dawn and dusk, respectively, at 6 a.m. and 6 p.m., until drought-stress induction and first time-snap (TS-1) on 22 August 2018. The first-time snap imaging was considered internal control data where both control and drought-affected plants were understood at par. Afterward, irrigation in three replicated pots was withheld for fifteen days until 6 September 2018 (TS4) for drought treatment, and the other set was well watered until maturity, as a control treatment. The varying levels of drought-stress were monitored to be around 12% SMC, i.e., half the soil moisture saturation level. After drought treatment, drought-affected plants were fully recovered with life-saving irrigation water up to 25% SMC. The final time snap of image data was acquired five days after recovery.

To avoid evaporation water loss, 500 g of solid, ultra-low weight, white-colored LDPE polypropylene beads were applied just above the soil surface of each pot as a mulching layer up to 3–5 cm in height. Therefore, the above ground-biomass of the plant was open to the environment for transpiration. We measured the transpiration by the gravimetric weight loss of the pot before and after watering each pot using an automatic watering and weighing system.

The entire growing condition was a controlled environment with a precise environmental monitoring system for temperature, relative humidity, vapor pressure, CO_2_, and light. Irrigation was applied to each plant in each pot in a fully automatic and controlled method. The weather conditions of each greenhouse were measured with fully automatic sensors, with a data logging frequency of 1 min. 

For the entire experiment, the range of the weather was set manually as per standard. The temperature (32 °C daytime and 26 °C night-time sinusoidal), relative humidity (60%), amount of irrigation water, CO_2_ concentration (450 ppm), and average vapor pressure deficit (2.5 for daytime and 1.5 for night-time), were recorded to understand the physiological behaviors of the plant. The dataset acquired from the 2018 Genome-Wide Association study in 2018 for the rice experiment was ‘IARI RICE GWAS 2018’. Additionally, greenhouse environmental variables were controlled to ensure uniform growing conditions for efficient evaluation.

### 2.4. Destructive Measurement of Biomass and Leaf Area

Each plant’s biomass (BM) was recorded destructively to ground-truth the phenotype’s response against induced stress. Much research confirms biomass as one of the crucial traits for interpreting plant growth, vigor, yield, stress response, and for analyzing associated QTL [21,23,34,35]. The biomass information in rice is associated with transpiration and leaf photosynthesis [36]. Further, this trait is associated with a healthier number of leaves, the leaf area, number of tillers, and grain-bearing productive tillers per plant [21]. Therefore, phenotypes with high biomass and higher water use efficiency are desirable for breeding programs [37]. The biomass (fresh and dried) weights of 95 phenotypes were measured with a weighing machine after cutting one of each plant replicated in drought and control treatments for each phenotype.

We measured the biomass (BM), i.e., fresh biomass weight (FBW), which includes the fresh stem weight (FStW) and fresh leaf weight (FLW). In this work, biomass refers to the above-ground biomass weight (AGBW), which is also often referred to as shoot weight (SW). All these parameters were recorded in both drought and controlled conditions. Additionally, root weight (RW) was recorded. These variables or parameters in controlled conditions were termed as ‘FBW_C for fresh biomass weight in control treatment’, ‘FRW_C for fresh root weight in controlled treatment’, ‘FSW_C for fresh shoot weight in controlled treatment’, ‘FStW_C for fresh stem weight in controlled treatment, ‘FLW_C for fresh leaf weight in controlled treatment’, and ‘FAGBM_C for fresh above-ground biomass weight in controlled treatment.’ Similar terminologies were used for drought treatment (condition). We also used the general terminology of ‘TBW for Total Biomass Weight’, ‘BM for ‘Biomass’, ‘SW for Shoot Weight’, ‘StW for Stem Weight’, and ‘LW is for Leaf Weight’.
Biomass Weight (BMW) = Stem Weight (StW) + Leaf Weight (LW)(1)
Stem Weight Percentage in Biomass = (StW/BM) × 100(2)
Leaf Weight Percentage in Biomass = (LW/FBM) × 100(3)

We also measured the leaf area (LA) and stem area (SA) in cm^2^ for controlled and drought treatment plants, using a leaf area meter obtained from LI-COR company, model LI-3100C.

### 2.5. Image-Based Plant Traits Estimation

Images acquired from high throughput plant phenotyping in a non-destructive manner are generally utilized for the estimation of a large number of plant traits and to monitor experiments [10,27,38,39,40]. It involves capturing morphological and biophysical traits either under laboratory (controlled) conditions, or field conditions with variable environments. Image analyses are performed on plant images to estimate various image-based traits for phenotyping. These image-based traits have been referred to as i-traits in a few recent research studies, and have been used for HTPP and GWAS studies [14]. Many phenomics studies have been carried out to link these phenotypes to genomic expression [10,39,40,41]. These traits also help to gain a detailed understanding of crops, such as their geometry and morphology.

In the current work, we aimed to estimate a few image traits associated with phenotypes to understand their variability, and to measure geometrical and physio-morphological characteristics. These traits were, namely, the number of leaves (NL), bounding box area (BBA), convex hull area (CHA), number of leaves per convex hull area (NLPCHA), plant aspect ratio (PAR), and plant elongation rate (PER). These parameters were estimated from the top view (such as NL, BBA, CHA, NLPCHA, PAR) and side view (such as PER) of the plant’s RGB images.

We aimed to establish the relationship between the various traits and understand the heterogeneity among the phenotypes. We clustered the genotypes into different groups according to their phenotypic characteristics to understand variability for breeding programs and the GWAS study. Such estimation of image-based traits had the additional purpose of dissecting the genotype–phenotype relation due to variation in environmental factors. This offered a platform to find new genes associated with particular phenotypic characteristics. 

To calculate the image-based traits and analyze the HTPP-arising big-data, various methods, principles, and algorithms were borrowed from image processing, computer vision, statistical pattern recognition, machine learning, and deep learning [29,42,43]. We employed YOLO [31,44] a deep learning algorithm for leaf detection and, thus, leaf counting, in our work, and employed other computer vision for estimation of the different traits [45], as shown in Figure 3.

#### 2.5.1. Counting the Number of Leaves per Plant in Rice

The number of leaves per plant in rice is a crucial trait. Many studies have found that the number of leaves, shoot dry weight, and the number of tillers were essential characteristics associated with the growth of rice plants [21,23,46,47,48]. Moreover, the leaf number is also an essential parameter for understanding the developmental growth stages and the phenology of rice in different genotypes [46,47,48]. It is a more crucial physio–morphological trait associated with a plant. This trait is required to be estimated because it is also an indicator of plant vigor, plant health status, and the developmental stages of a plant, and is associated with leaf area index (LAI) and stress condition [36,37,48,49]. It directly contributes to plant biomass in rice crops. It is mainly the healthier and broad leaves that contribute more to biomass. Thinner leaves generally contribute less to biomass and the formation of economic grain yield per plant. In this experiment, we estimated the number of the leaves for each rice genotype in control and drought conditions, at the different growth stages, using images.

Furthermore, we explored the association between plant height and number of leaves, using the acquired rice plant images, offering understanding of the emergence of newer leaves. Several studies have confirmed the association between plant height and number of leaves, with biomass, in different crops [36,48,49,50]. We estimated different heights from the side view of the image, as an i-trait at different drought stages, and corresponding to the number of leaves.

Recently, a large amount of research work has been conducted and applied in agriculture for prediction, object detection, and classification tasks employing AI-ML and deep learning (DL) in supervised and unsupervised learning, owing to its intelligent systems, speed and accuracy, for smart agriculture (SA), precision agriculture (PA) and high throughput plant phenotyping [9,42,51,52,53,54,55,56,57]. Few studies have yet been reported on leaf count in rice using images, compared with other plants such as maize and rosette [58,59]. Generally, the leaves of these (maize, rosette) plants are broad and have sparse distribution, making counting tasks easier than for rice. Rice at the vegetative or reproductive stage has an increased complexity and overlapping of rice leaves, thus making counting a challenging task. It becomes challenging to count hidden leaves near sheath areas, just above the ground, as shown in Figure 4. Therefore, we tried to count the visible leaf tip, with a hypothesis that “each leaf has a single tip in the rice plant”.

Further, by counting tips, the associated leaves can be calculated, i.e., the number of tips is equal to the number of leaves. In computer vision, the objects visible to the human eye are visible to the camera. Therefore, counting the hidden leaves becomes a tedious task in a rice plant, especially in the occluded area.

Leaf counting to extract the leaf features of maize plants was conducted by a convolution neural network (CNN) [58]. The CNN model was motivated by Google Inception Net V3, which used multi-scale convolution kernels in one convolution layer. To compress feature maps generated from some middle layers in the CNN, the Fisher vector (FV) reduced redundant information. Finally, these encoded feature maps regressed the leaf numbers by using random forests [58]. This algorithm achieved a mean square error (MSE) of 0.32 in a single maize data set.

We trained the YOLO-V3 model for leaf count. YOLO-V3 is a single CNN model with 75 convolutional layers. This model used the CNN network to predict the bounding boxes around an object of interest and with class probabilities of the bounding boxes from the images. YOLO predicted these boxes’ bounding boxes and class probabilities directly from the entire image using a single convolution neural network. In YOLO, object detection is a regression problem. This algorithm synergizes two tasks, object classification and object localization. One task locates the object and classifies it into different classes in object detection. Generally, localization is performed by drawing bounding boxes around the object. These bounding boxes contain the center point coordinates, width, and height, as in our label leaf tip, Figure 5 and Figure 6. To forecast bounding boxes, YOLO-V3 uses dimension clusters as an anchor [31,44]. Where the network predicts the four bounding box coordinates, such as, in this study, around the leaf tip, it is known as: t_x_, t_y_, t_w_, t_h_. If the cell is offset from the top left corner of the image by (c_x_, c_y_) and the bounding box prior has width and height, p_w_, p_h_, then the predictions correspond to:b_x_ = σ(t_x_) + c_x_
b_y_ = σ(t_y_) + c_y_
$$b_{w}=p_{w}e^{t_{w}}$$
$$b_{h}=p_{h}e^{t_{h}}$$

The sum of squared loss is used by this network during training. In the current work, only one class, the tip of a leaf, was considered and used as input for the network with its four coordinates, therefore, our case was detection and localization, not classification. We predicted that leaf tips were associated with corresponding leaves in a rice plant.

In this study, we labeled nearly twenty-five thousand leaf tips from 750 images of rice plants. We used only one class of objects, the leaf tip (shown in Figure 6a). We labeled leaf tips with a bounding box around them, Figure 6b. A deep learning model requires training using labeled data in supervised learning, especially for visual object detection. We labeled our image data for leaf tip with LeafTipMarker software, developed by us for annotation, as shown in Figure 5. Each annotated leaf tip had a bounding box coordinate. The tip was centered in the bounding box, Figure 6c, for the correct annotation and better prediction result of the leaf tip. This also helped to achieve a higher IoU of the predicted leaf tip. However, YOLO can be used for multiple classes of objects for images and video frames. It is commonly employed in an array of applications for object detection owing to its speed and accuracy [31,60,61]. We trained YOLO and performed hyperparameter tuning for an input size of 608 × 608, and anchor boxes (2 × 2, 10 × 10, 16 × 16, 20 × 20), to converge the model and for better detection accuracy. We proposed a framework, and the schematic workflow of our study is shown previously in Figure 3.

**Figure 5 plants-11-02663-f005:**
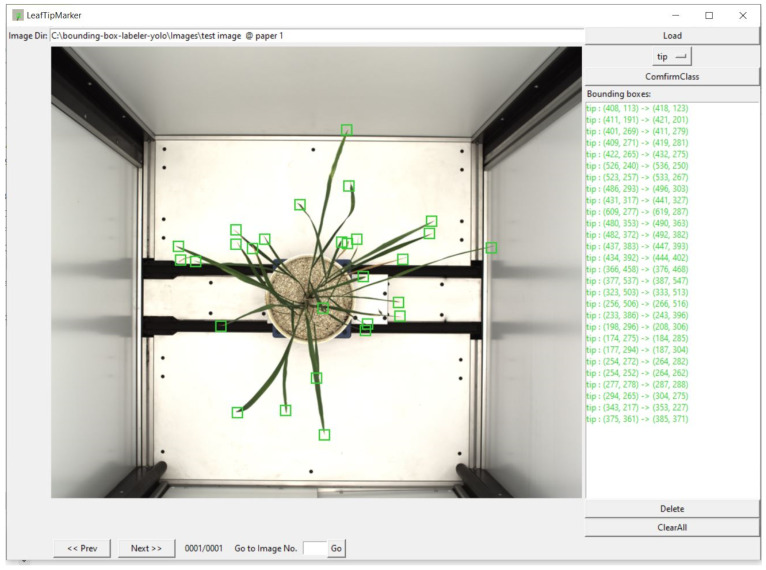
Annotation of leaf tips in a rice plant, using LeafTipMarker software.

#### 2.5.2. Convex Hull Area (CHA)

Convex hull is an essential geometrical property of an object in image processing [62]. It is associated with the shape of an object and can be used for image classification, shape detection, and various image-based applications related to it. In plant phenotyping it is related to the plant’s shape and structure. Therefore, this geometric trait can be utilized to understand a genotype, classify genotypes, and is employed in various stress studies and drought-specific genotype behavior [63]. Generally, when plants are under stress (drought), the total convex hull area becomes reduced for spread-type cereal and grass crops, and the convex hull increases for erect type plants [63]. However, convex hulls reduce for all types of genotype at the permanent wilting point or advanced stage of drought-stress due to the complete shrinkage of the plant [63].

For rice crops, this trait is critical. We can make inferences about a plant with this trait, and quantify the differences between drought and control (well-watered) plants. Further, we considered this trait to understand the genotypes, i.e., erect, spread type, or open type genotypes, based on the images’ top view or side views.

In the present study, we calculated the convex hull from the predicted tip coordinate of the leaf tip. As the leaf tip bounding box is tiny in our research, 10 × 10 pixels, we utilized coordinates innovatively to estimate the convex hull, Figure 6c,d. We aimed to keep the tip in the center while labeling the leaf tips with the ground truth bounding box. This bounding box coordinate included the center coordinate of the bounding box, which had the additional benefit of using these coordinates for convex hull estimation, Figure 6c,d.

**Figure 6 plants-11-02663-f006:**
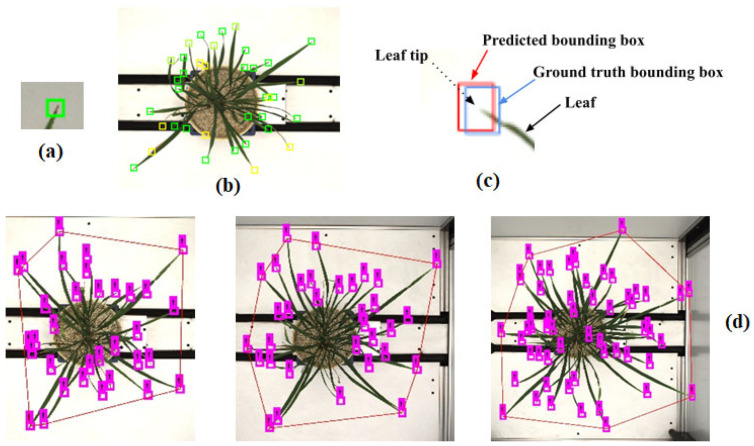
(**a**) Leaf tip, bounding box annotation for the leaf tip, (**b**) labelling of the leaf tips in a rice plant, (**c**) leaf tip at the center of the bounding box coordinates, (**d**) temporal increase in the number of leaves, and thus convex hull area (CHA); the number of leaves per convex hull area is a potential ‘temporal trait’ to understand the plant type, growth and stress behavior.

#### 2.5.3. The Number of Leaves per Convex Hull Area

We proposed this trait, the number of leaves to its convex hull area, as morphological and temporal traits [33]. This trait contains information about the plant architecture and growth pattern, as shown in Figure 6d. It indicates the leaf spread pattern, particularly in rice. A smaller number of leaves with a smaller convex hull area will be more erect and denser, than a lesser number of leaves with a larger convex hull area, known as a spread-type plant. This trait can also be used for classification and plant health monitoring.

#### 2.5.4. Plant Aspect Ratio (PAR)

Plant spread is easily visualized with PAR; for example, if we consider the top view of the plant, so PAR is the ratio of the plant width to length (height). This trait is also associated with the architecture of the rice canopy. The plant spread of equal length and width will be erect and to semi-erect. If, for the same PAR, the number of leaves increases, then the plant will be more compact and denser. This can determine how genotype plants spread in a controlled environment, and will help to understand planting density in the field.

#### 2.5.5. Bounding Box Area (BBA)

A bounding box is a hypothetical rectangle around an object that serves as a point of reference for object detection and creates a collision box for that object. The area encompassed by a Bounding Box is Bounding Box Area. Using the length and width of the plant, the bonding box area is estimated for each rice plant. This is a crucial temporal geometrical trait for obtaining the total spread area of the plant. It is a trajectory-based trait as BBA increases with the plant’s growth.

### 2.6. A Framework for Drought Quantification Based on the Number of Leaves

In conventional phenotyping, the drought treatment/waterlogged treatment is assessed based on a visual scale of high shoot growth with many leaves and tillers, at a level of ‘5’ or ‘10’ (depending upon whether the scale is up to 5 or 10), to the lower end of the scale at level ‘1’, with low shoot growth, and a low number of leaves and tillers, thus resulting in low biomass. In general, these indices are either a relative parameter or based on normalization, and are considered for quantification and classification [64,65,66]. These various stress or tolerance indices— stress tolerance index (STI), Stress Susceptibility Index (SSI)— have been developed to quantify stress based on relative growth or change in biomass, due to stress (drought), or non-stress (without drought) conditions.

Similarly, we conceived and proposed this framework and estimated an index such as stress tolerance index (STI) based on the leaf emergences rate (NER) in control and drought conditions during peak drought stress, since the start of the drought. It is the ratio of the number of leaves which set in the rice plant during peak drought conditions, to control conditions. The total number of leaves and leaf emergence contribute significantly in the biomass of the rice plant.
(4)$${\left(\mathrm{Stress}\;\mathrm{Tolerance}\;\mathrm{Index}\right)}_{\;\mathrm{Leaves}\;\mathrm{Emergence}\;\mathrm{Rate}}={\left(\mathrm{STI}\right)}_{\mathrm{LER}}=\frac{\Delta nl_{i,j}^{D}}{\Delta nl_{i,j}^{C}}\times 100$$
where, $\Delta nl_{i,j}^{D}$ is the number of leaves emerging for *i*^th^ genotype during *j*^th^ time period in drought conditions, and $\Delta nl_{i,j}^{C}$ is the number of leaves emerging for *i*^th^ genotype during j^th^ time period in controlled conditions.

We calculated the ratio of the number of leaves which emerged in a genotype during peak drought time (TS4, time snap 4) to the initial day of drought-induced (water-limited) condition (TS1, time snap1), to, the number of leaves which emerged in a genotype in the control (well-watered) condition for the same period of peak drought time (TS4, time snap 4) to the initial day of drought-induced condition (TS1, time snap1). This gave the value of LER; a higher value of LER meant that the genotype was more drought-tolerant or adaptive because a higher number of leaf settings were observed in drought conditions, compared with the control condition. We scaled its value between 0 and 100%. A result of more than 75% was considered drought-tolerant or adaptive. A lower value meant the number of leaf settings was less during the drought-stress period as the genotype was drought-susceptible. This was a straightforward indicator for identifying and clustering the genotype for screening drought-susceptible and adaptive genotypes.

### 2.7. Statistical Analysis

Statistical analysis of the data was performed using R 4.1.0 software and its different packages. We used R packages to plot the dendrogram. At the same time, we used Python packages, and Origin 2020b software to perform initial exploratory data analysis and plot the data. We used the Jupyter Notebook, an open-source web application for running Python codes, and different packages for the estimation of traits.

## 3. Results and Discussion

In this section, we present the results of our experiment, the framework for phenotyping, and clustering of the genotypes. We performed exploratory data analysis (EDA) on all samples, acquired destructively for genotypes, for different traits, and estimated image-based characteristics in a non-destructive manner. Table 1 and Table 2 list the descriptive statistics of these traits for the genotypes. In destructive sampling, 95 genotypes were considered for drought and control conditions. The traits, such as above ground biomass mass (herein referred to as biomass), leaf area, leaf weight, stem area, stem weight, and root weight, were recorded. The coefficient of variation (CV) for all features was quite good, showing variability among the genotypes; all traits had more than 20% CV. A higher value of CV ensures the diversity of the gene pool, which is better for GWAS and breeding programs.

### 3.1. The Rice Leaves and Their Contribution to Biomass

Various reports confirm the role of rice leaves in photosynthesis and contribution of biomass. Therefore, the percentage of leaf weight in fresh biomass weight was estimated. Figure 7a,b shows the scenarios of above-ground biomass in control and drought conditions for the genotypes, and the contribution of rice stem and rice leaf in the fresh biomass i.e., above-ground biomass, in destructive samples. It was found that rice leaves contributed, on average, to around 47% of the total biomass. However, for our ground truth data we observed that the value ranged between 30 and 70% for fresh and dry weight. Therefore, it can be inferred that rice leaves significantly contributed to the biomass for fresh and dry weight. This supported our leaf-counting hypothesis for the rice genotype discrimination on the basis of number of leaves: the higher the number of leaves, the more biomass.

Moreover, it was also found that AGBM was a better trait to compare and understand the genotypes for controlled and drought conditions. For specific genotypes, the value of above-ground biomass in drought conditions was slightly higher than in controlled conditions. This variation may be due to seed vigor, soil fertility, or the performance of the upland genotype. While analyzing these identical genotypes with AGBM, the control conditions had either higher values or were nearly at par. For genotypes (e.g., NAGINA 22, DULAR, EYPO, MOROBEREKAN) which are upland or tolerant to drought, the biomass differences in drought conditions as compared to control were insignificant, in contrast to drought-sensitive genotypes.

### 3.2. Estimation of the Number of Leaves for Each Genotype in Controlled and Drought Conditions and Quantification of Drought Response

To estimate the number of leaves in each rice genotype, we trained the YOLO-V3 model from scratch. Around 25,000 leaf tips were labeled from 750 RGB images of rice plant. We increased our train-and-test set from the previous work [30]. However, we split the train, test and validation set in the same ratio of 80:10:10. We used the top view of the rice plant for counting, with a 6576 × 4384 resolution. For training purposes, we resized the images to 3700 × 3700. Before inputting the images to the YOLO model, we resized them to 608 × 608. We aimed to train the model to resize images by reference to the architectural limit. We used an NVIDIA GPU, GeForce GTX 1080 Ti 11GB graphic card, for training the model.

The YOLO deep learning model was adopted to detect leaf tips and corresponding leaves for a rice plant, as shown in Figure 5. We trained the model with an image batch size of 16 and tried various anchor box sizes: 2 × 2, 8 × 8, 10 × 10, 16 × 16, 20 × 20, and 32 × 32, to obtain the optimized result of leaf tip detection. We obtained the best outcome using the 16 × 16 and 20 × 20 size anchor boxes, depending on our input training image size of 608 × 608 for the YOLO pre-trained model. For labeling the leaf tip, a ground truth bounding box of size 16 × 16, 14 × 14, 13 × 14, 13 × 13, or 10 × 10, was tried. The predicted bounding boxes around the leaf tips were in the same range, see Figure 6. YOLO generally uses a multiclass object, yet we considered only one class, i.e., leaf tip, for object detection. We trained the YOLO model on training data for different epochs and iterations to converge the model. We presented the leaf tip detection result in Figure 8 for a genotype at 100, 1 K, 10 K, 30 K, 50 K, and 56 K iterations; the model converged with better detection at 56 K iterations.

We started our model training with the initial learning rate set to 0.001 with a decay of 0.0005 applied at iterations of 40 K and 45 K, and the batch size of the images, of 16. We used stochastic gradient descent optimizer for the training. YOLO performed exceptionally well for detecting leaves, especially for sparse and uncrowded leaves and their tips. Due to the complexity of the leaves near the culm area in the rice, which is very crowded and dense, see Figure 4b, the accuracy went down. However, all healthy and broad leaves were detected and counted in. The model performed very well for the small and sparse plants, where detection accuracy was up to 98%, and for dense plant types, where accuracy was up to 86–92%. We achieved better results than our previous model [30] due to an ample training and testing set of the sample data.

Depending on the leaf distribution of plants and their complexity, a different level of accuracy was achieved in the different plants chosen for leaf counting. Generally, it ranged from 86% in very dense and crowded rice plants, to 98% in very sparsely distributed rice plants.

### 3.3. A Novel Procedure for Working with YOLO-V3 for Small Fixed-Sized Squared Boxes

We note that leaf-tip counting has unique characteristics that are different from generic object detection tasks. First, our study involved small-sized boxes. Second, all the boxes were of fixed length and equal in size. Third, leaf tips are not simple objects with fixed boundaries, such as those used in object detection problems. Lastly, we are interested in counting leaf tips, and the tip area is not large.

We initially observed that our experiments did not converge if we directly used YOLO-V3. We resized the input of YOLO-V3 to 608 × 608. Each annotated leaf tip corresponded to a resolution of 608 × 608, a square of length of 16. Next, we observed that our model did not converge for leaf tips if we used anchor boxes calculated using k-means or fixed them to squares of size 16 each, perhaps because leaf tips are small in size. To resolve this, we tried increasing the size of the leaf tips in the ground truth, but it led to many overlaps among different boxes, making the learning difficult. Next, we reduced the size of anchor boxes to make the ground truth boxes relatively bigger (than the anchor boxes). We carried out three more controlled experiments with fixed square anchor boxes of 10 × 10, 6 × 6, and 2 × 2 pixels.

Interestingly, our experiments started to converge with anchor boxes sized 2 × 2 pixels each. We believe that our procedure can be helpful in other similar applications which involve small squared boxes. Figure 8 shows the predictions of YOLO-V3 trained with our procedure on 2 × 2 anchor boxes; the model interestingly toggles between optimizing the number of leaf tips and the length of the squared-sized tip. For the initial 30 K iterations, the model improved on tip size and count. For iterations from 30 K to 56 K, we observed that count accuracy improved. The model overfit training data for subsequent iterations and predicted many false positives. We chose the model with the optimal number of leaf tips (56 K iterations) for our task, ensuring that each count of predicted leaf tips had some overlap with the corresponding ground truth annotation.

We counted the entire leaves in the plant by counting the total number of corresponding leaf tips, as rice plant has a single tip for each leaf. We observed that the number of leaves in drought-sensitive crops was less in drought conditions than in well-watered conditions. We asserted this at the higher *p*-value of 0.5 and 0.01. There was a significant difference between the number of leaves setting in many droughts’ sensitive genotypes. However, for drought-tolerant genotypes, the leaf count (leaf setting) was at par; no significant difference was observed at a *p*-value of 0.01. We inferred that leaf number is a crucial trait in rice, equal to another image-based trait. With this trait, we quantified the drought-tolerant and drought-sensitive genotypes, and it was evenly relative to different growth stages. We observed a significant change in the number of leaf settings after inducing drought-stress in identical genotypes. This can be further used to monitor plant growth at different stages, Figure 9.

### 3.4. Correlation Analysis of the Traits Based on Destructive Sampling Observations and Image Traits under Drought and Control Response

We correlated the destructive data of the fresh biomass weight, shoot weight, root weight, leaf weight, leaf area, shoot area, and total area. These traits are additive and were correlated with each other. Figure 10 shows the correlation coefficient of these variables and the heat map in control treatment and drought conditions (treatment). We observed a strong correlation between the above-ground biomass and leaf weight, with a correlation coefficient of 0.91 in controlled conditions, and 0.93 in drought conditions. Similarly, for the total biomass and leaf weight, the correlation coefficient was 0.93 in drought and approx. 0.93 in controlled condition. A similar trend was observed for fresh root weight and biomass, shoot weight and biomass, root weight and above-ground biomass, shoot weight and above-ground biomass, and leaf weight with above-ground biomass in controlled and drought conditions. Except for root weight and above-ground biomass, and root weight and shoot weight, a very high positive correlation coefficient was found, of more than 0.75.

**Figure 10 plants-11-02663-f010:**
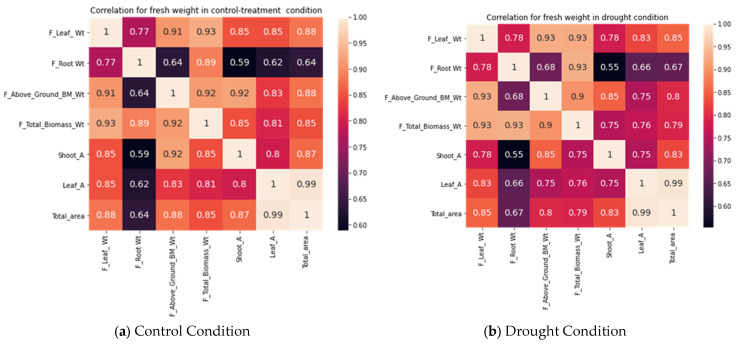
Correlation among different quantitative traits in (**a**) control or well-watered condition and (**b**) drought treatment for rice plant.

**Table 1 plants-11-02663-t001:** Descriptive statistics of the quantitative traits and image traits of different genotypes for plants in control treatment.

	T_BM_Wt (g)	A_G_BM_Wt (g)	S_Wt (g)	R_Wt (g)	L_Wt (g)	LA (cm^2^)	SA (cm^2^)	TA (cm^2^)	NL_i_	CHA_i_ (cm^2^)
Min	11.30	6.30	2.20	5.00	4.10	205.30	34.20	277.30	12	442.60
Max	138.25	83.08	61.21	75.60	25.98	1331.50	320.00	1602.30	106	12,833.12
Mean	70.80	33.13	18.34	37.66	14.79	662.82	139.92	802.75	42	6230.512
STD	25.53	15.51	10.92	13.11	5.41	272.95	74.23	333.86	24	1432.11
CV	36.05	46.82	59.58	34.82	36.58	41.18	53.06	41.59	57.14	22.98

T_BM_Wt—total biomass weight; A_G_BM_Wt—above ground biomass weight; S_Wt—shoot weight; R_Wt—root weight; L_Wt—leaf weight; LA—leaf area, estimated from destructive sample by leaf area meter; SA—shoot area; TA—total area of rice plant; NL_i_—number of leaves, estimated from image; CHA_i_—convex hull area, estimated from image.

**Table 2 plants-11-02663-t002:** Descriptive statistics of the quantitative traits and image traits of different genotypes for drought condition.

	T_BM_Wt (g)	A_G_BM_Wt (g)	S_Wt (g)	R_Wt (g)	L_Wt (g)	LA (cm^2^)	SA (cm^2^)	TA (cm^2^)	NL_i_	CHA_i_ (cm^2^)
Min	6.76	2.90	1.50	3.06	1.40	45.30	12.50	57.80	8	324.48
Max	115.77	59.33	37.53	69.88	23.81	1320	281.60	1589.00	84	10,833.21
Mean	65.15	28.85	15.57	36.30	13.28	573.14	131.75	704.89	38	5640.32
STD	22.87	11.30	7.10	13.52	4.84	261.05	58.84	307.47	18	1188.20
CV	35.11	39.18	45.57	37.24	36.45	45.55	44.66	43.62	48.36	21.0661

T_BM_Wt—total biomass weight; A_G_BM_Wt—above ground biomass weight; S_Wt—shoot weight; R_Wt—root weight; L_Wt—leaf weight; LA—leaf area, estimated from destructive sample by leaf area meter; SA—shoot area; TA—total area of rice plant; NL_i_—number of leaves, estimated from image; CHA_i_—convex hull area, estimated from image.

Biologically, rice leaf and shoot contribute significantly to total biomass (for fresh weight as well as for dried weight. More leaves directly contribute to biomass and economic yield [16]. Considering this, we also correlated the number of leaves and leaf weight. A positive correlation was observed between the number of leaves and leaf weight. The number of leaves was estimated from the RGB images of each rice genotype. Here, we considered the top view of the plant.

We also checked the correlation between the number of leaves and the convex hull area of each plant. This correlation is associated with the plant type. A larger number of leaves with a small convex hull confirmed the plant as being an erect to semi-erect plant type and a smaller number of leaves with a higher convex hull area confirmed it as being a spread plant type. However, a larger number of leaves with a larger convex hull area also confirmed the plant as belonging to the spread plant type. A decrease in convex hull or bounding box area was observed and also associated remarkably with leaf rolling or stress in genotype [67]. Figure 11 depicts the leaf number count in one of the drought-sensitive genotypes, which affected the total biomass and leaf weight.

### 3.5. Convex Hull, Convex Hull Area, Bounding Box (Bounding Rectangle), Plant Spread Calculation, and Correlation with the Number of Leaves in Rice Genotypes

Plant size was well explained with the convex hull, convex hull area, bounding box, and plant spread, as these geometrical properties are associated with biomass. Figure 12 depicts the various geometrical properties of a rice plant, estimated with image processing and computer vision algorithms. We found a significantly positive correlation between these traits, see Figure 13.

### 3.6. Correlation with the Number of Leaves to Leaf Weight of the Plant

After estimating the number of leaves, we aimed to determine the correlation between the number of leaves and leaf weight, as we observed in destructive samples that leaf weight was significantly contributing to the total biomass and above-ground biomass in the rice plant. Table 3 presents the correlation value between the leaf number, leaf weight, and convex hull which was positive and highly corelated. 

**Figure 11 plants-11-02663-f011:**
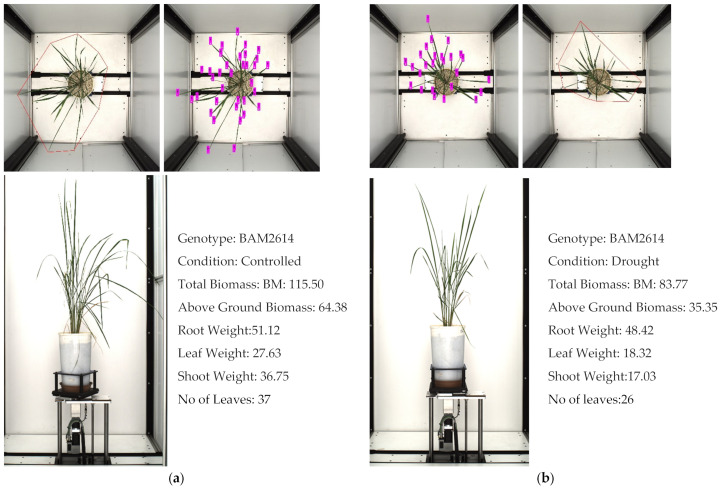
The number of leaves is an important trait associated with biomass in rice. The images show one of the rice phenotypes, BAM 2614, in: (**a**) control treatment, and (**b**) drought treatment.

We observed the correlation between the leaf weight and the number of leaves was positive, at 0.91 and 0.90 in controlled and drought treatment, respectively. Similarly, we witnessed a significant correlation between the leaf number and convex hull of the plant, at 0.83 and 0.86. We can infer from this that the leaf weight and biomass are directly associated with the increase in the number of leaves.

**Figure 12 plants-11-02663-f012:**
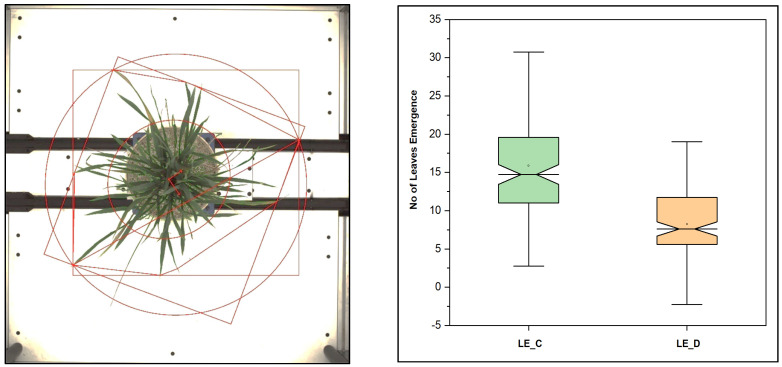
Different parameters, geometrical traits in the rice plant estimated from the top view, and leaf emergence in rice in control and stress, drought condition. More leaf emergence in control conditions compared to drought conditions was observed.

### 3.7. The Number of Leaves, Number of Leaves per Convex Hull, and Other Geometrical Traits

By measuring the number of leaves, the number of leaves per convex hull area, and other geometrical traits, as shown in Figure 12, we assessed the condition of the plants in water stressed and well-watered conditions. A drought-sensitive genotype in healthy conditions had a higher number of leaves (NL), convex hull area (CHA), number of leaves per convex hull area (NLPCHA), bounding box area (BBA), and compactness, compared with drought-resistant genotypes. Depending upon the plant erectness and spread type, the CHA and NLPCHA could change; a report found that these two properties, CHA and BBA, may reduce at a significantly higher stage of stress with PAR [63].

For drought resistant genotypes such as DULAR, HEERA, Anjali, NERICAL-L-42, RASI, and A-DEY-SEL, it was observed that the number of leaves which emerged during the period of induced stress to the peak value of the stress (drought), the number of leaves was much higher than for drought-susceptible genotypes, such as BAM 1689, VANDANA, HONG, SATHI, ANANDA, SUWEON, BAM-1812, and PUSA-44 during the study period.

The convex hull area for these genotypes was also higher during the well-watered condition compared with in the stress condition. Generally, the higher convex hull area depended upon the plant spread type architecture. A few genotypes that are semi-erect types were observed to have higher convex hull areas than compact and erect genotypes. In the core set, the maximum genotype group was erect and semi-erect, compared with the open and spread genotypes.

Another essential trait was the plant spread type. It was previously found that with the increase in stress, t speediness increases for rice plants, which are erect and semi-erect types; however, at terminal stress or at higher stress levels, the value decreases [18,36,63]. We found similar trends for our core set of genotypes. These traits, based on RGB images, were significant at *** *p* < 0.05 and ** *p* < 0.001.

**Figure 13 plants-11-02663-f013:**
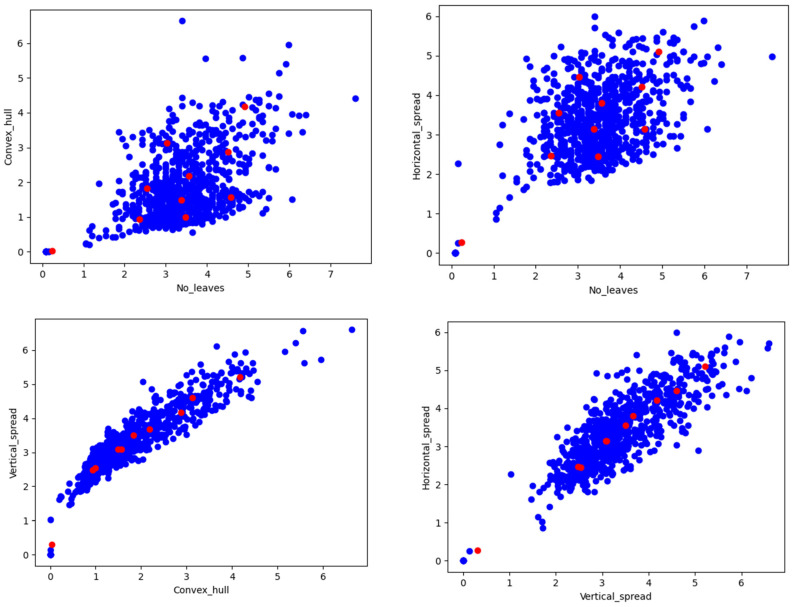
Scatter plot for the number of leaves vs. geometrical properties (convex hull and horizontal spread of the plant); and among geometrical properties of the plant.

### 3.8. Cluster Analysis of the Genotypes Based on the Different Traits

We plotted a radial dendrogram (Figure 14) to group the genotypes for drought-adaptive and drought-sensitive genotypes, based on the number of leaf counts and the number of leaf emergences in drought and well-watered conditions estimated as per Equation (4). The groups clearly showed that drought-tolerant genotypes such as DULAR, HEERA, Anjali, NERICAL-L-42, RASI, A-DEY-SEL, a few BAM series, RASI, ABHAYA X DAGADESI, Nagina fell into one group with a higher value of leaf emergence rate or the number of the leaves per plant after induced stress. These traits are associated with plant size, moderately associated with perimeter area ratio (PAR), leaf area, and above-ground biomass. The LER for these genotypes was more than 75%.

BLACKGORA, EYPO, WAY RAREM, BAM8364, BAM712 and BAM747 fell into another category, and were moderately tolerant to drought. A similar pattern was found in all traits, except for a few exceptions in above-ground biomass. Another group which was light tolerance, we found that MOROBEREKAN, A DAY SEL, BAM 3690, BAM 3181, PMK-2, and BAM4496 fell into a third category, with a LER of 65–85%.

The drought-sensitive group was comprised of BAM 1689, VANDANA, HONG, SATHI, AANANDA, SUWEON, BAM-1812, PUSA-44, and IRROEPYO. In this group, IRROEPYO, PUSA-44, SUWEON, and VANDANA are very well-known drought-sensitive varieties. For these genotypes, the LER was less than 60%. The primary category was the drought-sensitive genotypes, which comprised PUSA BASMATI 6, BAM2680, SAFRI-17, BAKAL, BAM 56, BAM 8364, APO, BAM 326, IR87707-445-B-B, and SATHI.

**Figure 14 plants-11-02663-f014:**
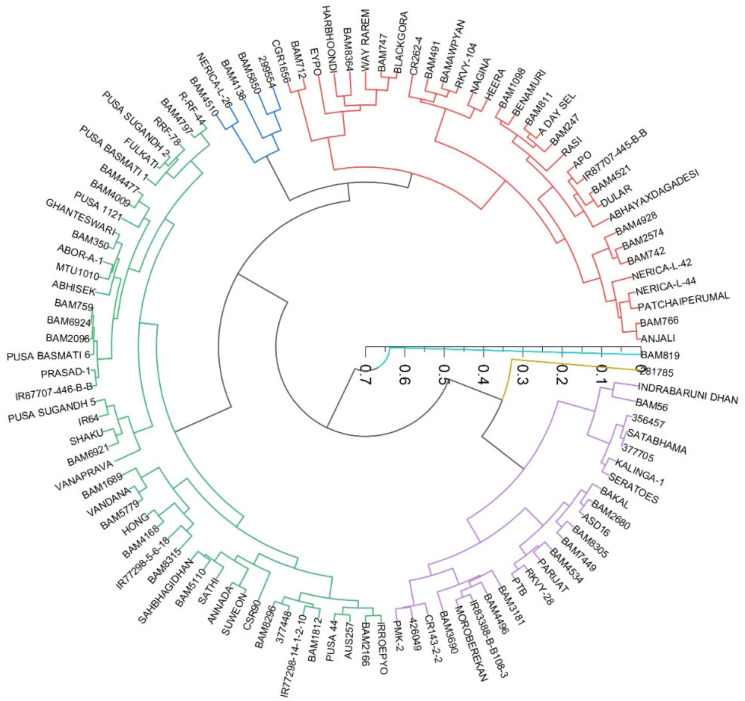
Grouping of genotypes based on the number of leaves, and the number of leaves that emerged between the initial days of induced stress and peak drought stress.

## 4. Conclusions

This paper demonstrated an application of deep learning and computer vision techniques to estimate image-based traits of rice plants for their different genotypes. We proposed a simple framework to discriminate between drought-tolerant and drought-sensitive genotypes based on the correlation between quantitative traits of the rice of destructive samples, and estimated image traits, notably the number of leaves. These image traits were observed from top view, as well as side view, images. It was observed that the number of leaves, a drought-adaptive trait associated with plant size and biomass, was significant enough to precisely distinguish for drought-tolerant genotype among 110 genotypes. All the results were found to be significant; therefore, we infer that image-based traits acquired from high throughput plant phenotyping have the potential for drought-specific selection of genotypes. Further work can be extended for the GWAS program to assist breeding and dissect the functional genomics.

## Figures and Tables

**Figure 1 plants-11-02663-f001:**
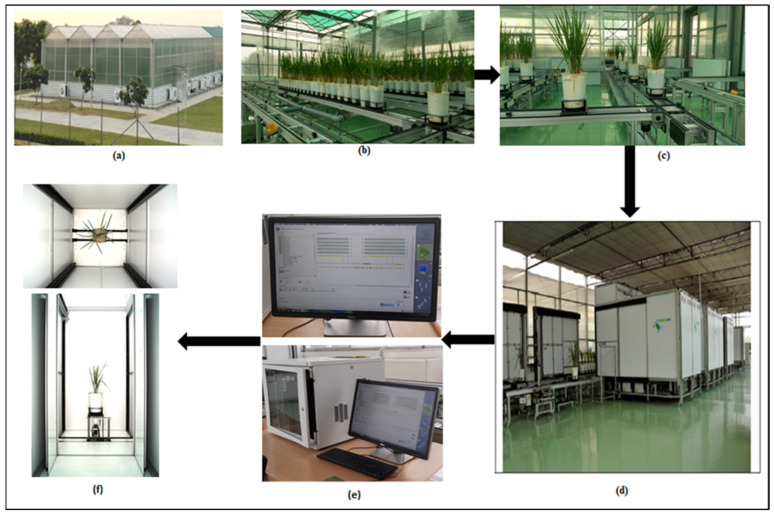
A plant-to-sensor type, high throughput plant phenotyping facility at Nanaji Deshmukh Plant Phenomics Centre (NDPPC), Indian Council of Agricultural Research–Indian Agricultural Research Institute, and their different components for the experimental setup, representing a general workflow: (**a**) NDPPC, external view; (**b**) rice plants in pots, transplanted inside the climate-controlled greenhouse; (**c**) rice plants are transported on carriages to the imaging chamber; (**d**) automatic imaging chambers, each chamber is equipped with a different sensor of interest, in this case, RGB; (**e**) monitoring and work station equipped with software; (**f**) image of a rice plant, top view and side view of the rice plant after imaging.

**Figure 3 plants-11-02663-f003:**
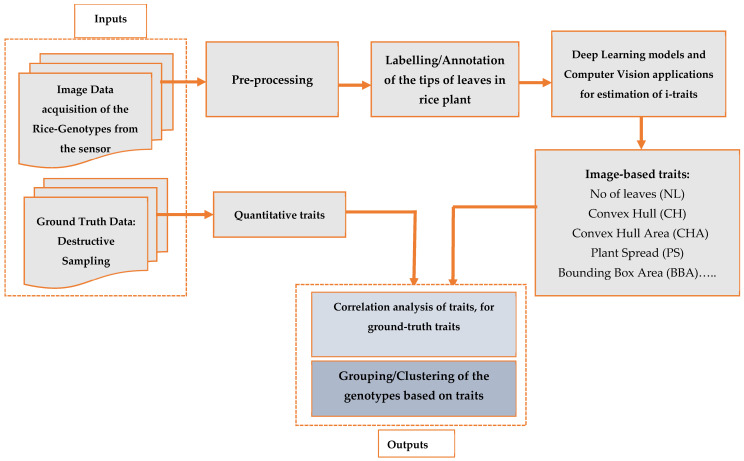
Schematic workflow of the framework for leaf counting, image-based traits estimations, and grouping of the genotypes.

**Figure 4 plants-11-02663-f004:**
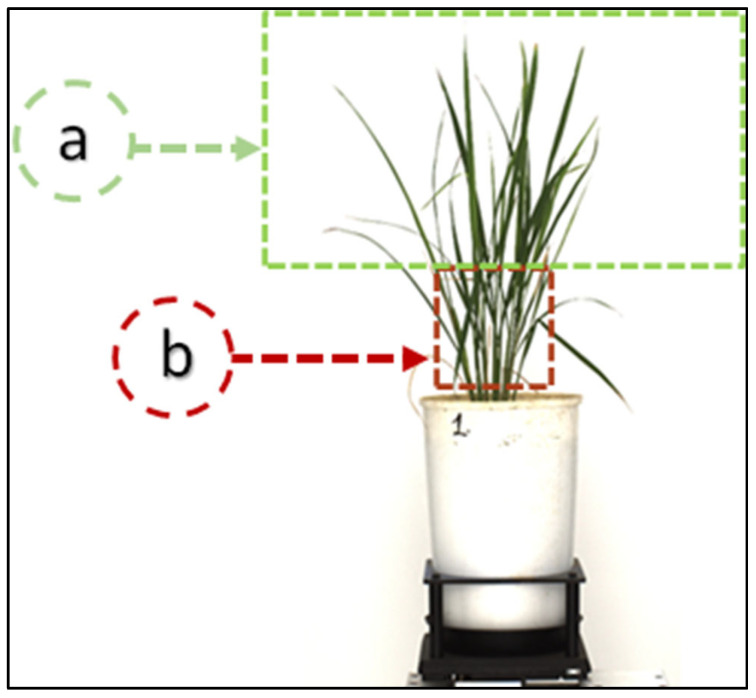
Rice plant at the vegetative stage depicting leaf distribution in a rice plant: (**a**) leaves are sparsely distributed and less overlapping at the upper part of the plants, whereas, (**b**) just above the ground, near the sheath area, the leaves are thin, generally hidden, and cluttered.

**Figure 7 plants-11-02663-f007:**
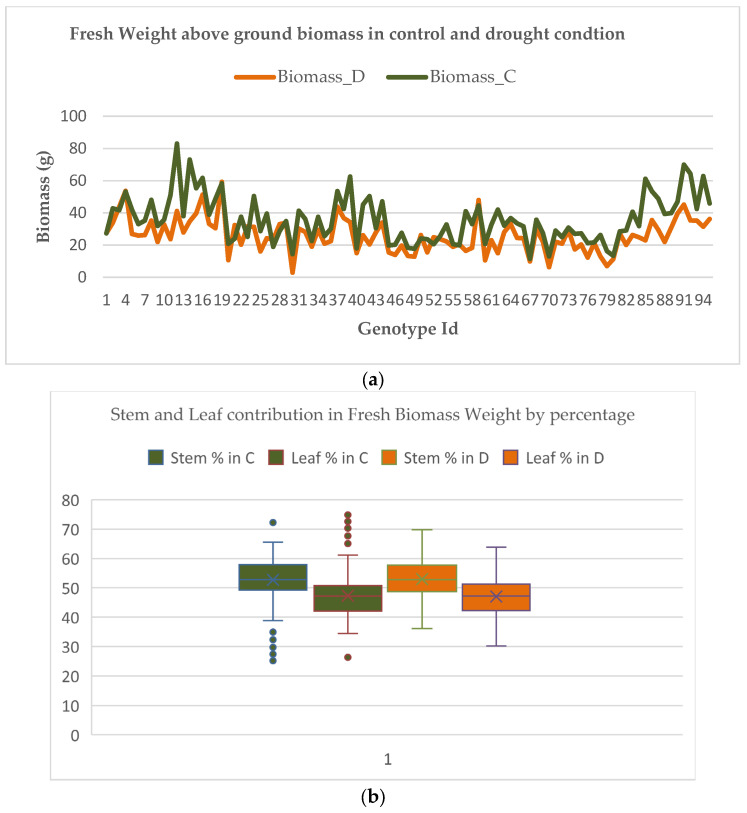
(**a**) Comparison of biomass (fresh weight) of the genotypes in drought and control conditions, and (**b**) stem and leaf contribution by percentage in biomass, for fresh weight.

**Figure 8 plants-11-02663-f008:**
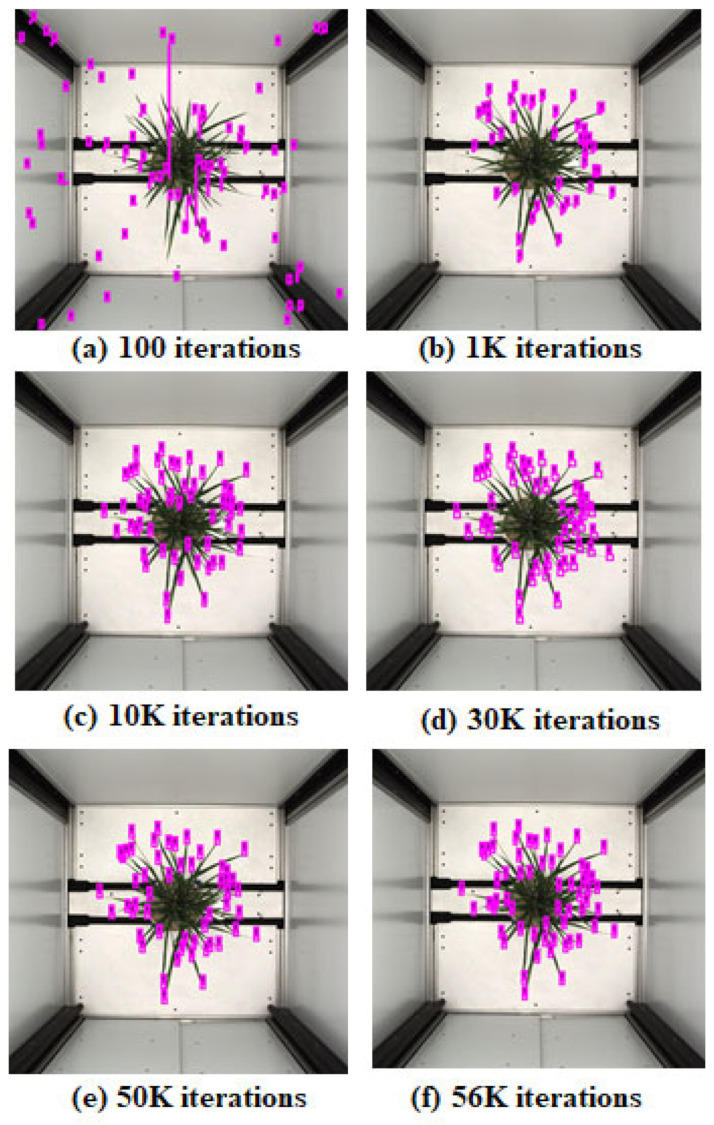
Training the model at different iterations of (**a**) 100, (**b**) 1K, (**c**) 10K, (**d**) 30K, (**e**) 50 K and (**f**) 56K iteration, and each iteration’s result for detection and prediction of leaf tip, a tiny object.

**Figure 9 plants-11-02663-f009:**
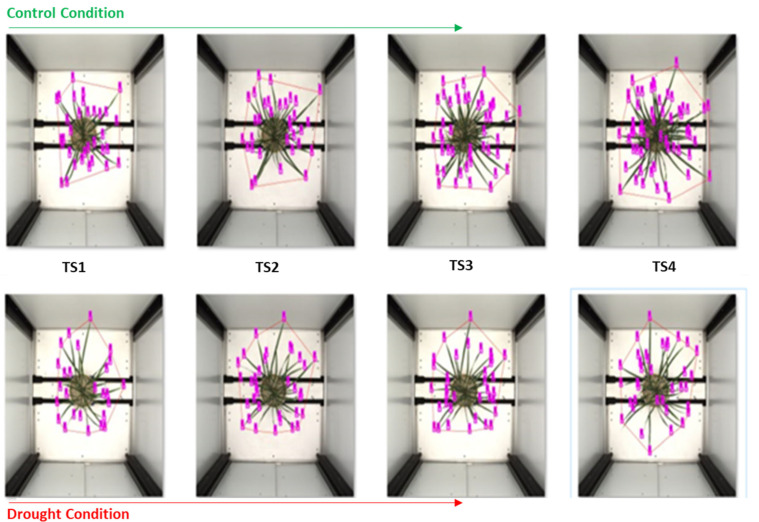
Convex hull and convex hull area estimation by predicting leaf tip and its coordinate without removing the background, and tip coordinates are used for convex hull estimation. Here, a genotype in drought and control conditions: in control, a large number of leaf settings are observed, compared with the stressed plant; also, no leaf rolling and a large convex hull area in the control plant is observed, compared with the drought-stressed plant with a smaller number of leaves, leaf rolling and comparatively less convex hull area.

**Table 3 plants-11-02663-t003:** Correlation analysis between quantitative traits and image traits.

Traits	Correlation
Leaf number and leaf weight (fresh weight control)	0.91
Convex hull and leaf weight (fresh weight control)	0.83
Leaf number and leaf weight (fresh weight drought)	0.90
Convex hull and leaf weight (fresh weight drought)	0.86

## Data Availability

The data are available from the corresponding author upon reasonable request.

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
