# Peer review of "Leaf Count Aided Novel Framework for Rice (Oryza sativa L.) Genotypes Discrimination in Phenomics: Leveraging Computer Vision and Deep Learning Applications"

_plants, 2022, doi:10.3390/plants11192663_

Round 1
Reviewer 1 Report
The manuscript titled “Leaf count aided novel framework for rice genotypes description in phenomics: leveraging computer vision and deep learning application” provides novel findings. However, two major concerns which needs to be addressed are:
1. Provide the well annotated sources codes freely to understand and replicate the methodology
2. Limited literature review search was done, many new studies were not cited or used for this study, please go through them, and cite appropriately.
Author Response
Dear Sir,
Please find the attached document, which is a response to your comments.

Reviewer 2 Report
This is the review report of the paper which is titled
“
Leaf Count Aided Novel Framework for Rice (Oryza-Sativa L.) Genotypes Discrimination in Phenomics: Leveraging Computer Vision and Deep Learning Applications “.
The paper is well written and presented. Good effort has been done, I have some minor comments
1- The title is too long; I suggest shortening it
2- Clear sentences about the research problem, in the other words, what is the problem of previous methods that this paper solved.
3- More details about the deep learning part, training parameters, feature visualization
4- Details about the used dataset with how it was divided for training, validation, testing, with ratio.
5- comparison with previous methods is necessary added.
6- More recent papers should be added such as
https://www.mdpi.com/2223-7747/9/10/1302
Author Response
Dear Sir,
Thank you for the comments and for evaluating our work, the response to your comments is in the attachment.

Reviewer 3 Report
Currently, agriculture consumes 75% of global water and that percentage could double in the next 50 years, if trends in population growth and current food production practices continue.
However, intensive agriculture with increased productivity to meet global food demands for an ever increasing human population creates serious environmental hazards, including drought. This hazard is exacerbated by climate change, and represent major threats to plant survival and agricultural productivity. Indeed, drought is commonly seen as the most important abiotic factor limiting plant growth and yield in many areas. In fact, drought is one of the most costly natural hazards in each year. With the world’s natural resources being depleted at an alarming rate, the race is on to find technologies that can ameliorate drought-related losses and also meet the requisites of economic, social, and environmental sustainability. Deep learning algorithms have emerged as an effective solution for many agricultural applications allowing farmers to take important decisions at the right time. They provide several advantages over the classical machine learning algorithms, such as their high ability to extract highly relevant features automatically. Also, recent advances in hardware and software technologies all along with the high availability of data make it possible to train and deploy such powerful techniques. Thus, the use of deep learning algorithms in crop classification tasks has greatly increased in the last few years, where we find several serious efforts to improve this task using recent deep learning algorithms.
The goal of this manuscript is to present and characteristics of computer vision and deep learning applications for the analysis of drought tolerant rice phenotypes.All parts of the manuscript is interesting and clearly summarize data. The authors have done a good job at describing the problem, the methods and the results.
GENERAL COMMENTS:
TITLE
The paper title is well stated, it is informative and concise.
ABSTRACT, INTRODUCTION
Abstract and Introduction were well written, and very good presenting the subject and research problem.
MATERIAL AND METHODS
Material and research methods are presented appropriately. Experimental setup and the description in the methods section are well structured.
RESULTS
The results obtained in this study are interesting. Results presented correctly.
DISCUSSION
In general, the discussion of results is correct and sufficient.
LITERATURE
The items of literature included in the paper are rather sufficient and adequate to the subject of the paper.
Please verify the correctness of the literature and make a linguistic correction of the text by native speaker.
Author Response
Dear Sir,
Thank you very much for appreciating our work. Please find the attachment having the response to your comment.

Round 2
Reviewer 1 Report
NA